# Nitrogen Enriched Organic Fertilizer (NEO) and Its Effect on Ryegrass Yield and Soil Fauna Feeding Activity under Controlled Conditions

Hesam Mousavi [1,2,*], Thomas Cottis [1], Gina Hoff [1] and Svein Øivind Solberg [1]

1   Department of Agricultural Sciences, Faculty of Applied Ecology and Biotechnology,
    Inland Norway University of Applied Sciences, 2318 Hamar, Norway; thomas.cottis@inn.no (T.C.);
    ginamhoff@gmail.com (G.H.); svein.solberg@inn.no (S.Ø.S.)
2   Gothenburg Global Biodiversity Center (GGBC), The University of Gothenburg,
    SE-405 30 Gothenburg, Sweden
*   Correspondence: hemousavi65@gmail.com

**Abstract:** This study aimed to investigate the effects of a new nitrogen-enriched organic-based fertilizer (NEO) on Italian ryegrass (*Lolium multiflorum* Lam.) yield and soil fauna feeding activity. Nitrogen is transformed from the air to manure by a plasma process. At the farm level, NEO could improve self-sufficiency and sustainability. The work was carried out under controlled conditions in two pot trials. Five fertilization regimes were used: no fertilizer, different amounts of mineral fertilizer, three NEO types, organic fertilizer (untreated manure), and organic fertilizer + different amounts of N in mineral fertilizer, including 14 treatments in trial one and 11 treatments in trial two. Besides evaluating dry matter yields, we utilized the Bait-lamina test system to assess the feeding activity of soil fauna. The results indicated a clear positive impact of nitrogen (N) on ryegrass yield where all fertilizers increased the yield in correspondence with their N availability regardless of the fertilizer type; whereas the yield was highest with mineral fertilizer up to our maximum level of 235 kg N ha$^{-1}$ in trial one and 175 kg N ha$^{-1}$ in trial two. The NEO fertilizers yielded in the same range as mineral fertilizers. The same clear pattern was not observed for soil fauna feeding activity. Instead, a tendency was observed where no fertilization tends to give the highest feeding activity. We saw no correlation between the yield and the soil fauna feeding activity. The feeding activity was highest in depth below 5 cm from the soil surface. Feeding activity also increased over time after fertilization. The NEO fertilizers had no more adverse effects on soil fauna feeding activity than other fertilizers. Other factors than fertilization alone are determining the soil fauna feeding activity.

**Keywords:** fertilization; nutrients; nitrogen; NEO; ryegrass; soil health; sustainability; yield

## 1. Introduction

Agriculture systems are extensively nourished with mineral fertilizers [1], and the global input of nitrogen (N) into the biological system is eight times higher today than it was in the 1960s [1–3]. Although fertilization has contributed to higher yields, the trade-offs are many and severe [3,4]. Therefore, sustainable use of fertilizers is of significant interest, and combining mineral and organic fertilizers or improving the efficiency of applied fertilizers are ways to maintain good yields and sustain soils [5]. The positive effect of N fertilizers on yield is documented well [6–8], but its effects on soil organisms and functions are more complex and often neglected [9,10]. Fertilization directly manipulates soil nutrients and stimulates alteration in soil functional communities, making the environment favorable for some functional groups and more unfavorable for others [11]. These functional groups are engaged in plants' well-being and ecosystem services in different ways, such as disease protection, pathogenicity, and nutrient turn-over [12]. Thus, identifying a fertilizer regime with the most negligible negative impact on soil functional organisms is of high priority and potentially leads to more sustainable agriculture.

Grasses are considered an important option for cover and forage crops in the Nordic conditions [13]. Perennial and annual ryegrasses (*Lolium* spp.) are among the most robust grasses in the region and are commonly used in agriculture fields and grasslands [14,15]. Nevertheless, having a fast growth rate, extended roots, high yields, high forage quality, and low tillage requirement, make annual ryegrasses a favorable crop in forage production and as rotation crops [16].

Nitrogen Enriched Organic fertilizer (NEO) is a newly patented organic-based fertilizer. Using electricity with a patented unit, nitrogen is transformed from the air to the manure by a plasma process. The manufacturer (N2 Applied, Asker, Norway) claims that the final product can be a suitable alternative for conventional organic and mineral fertilizers [17–19]. NEO enables end-users to produce their fertilizer locally and provides self-sufficiency, improving plant production sustainability [17]. NEO holds the same characteristics as cattle slurry but contains more nitrite and nitrate, has lower pH, and is more fluid due to filtration in the production process. As a result, NEO is claimed to have considerably lower emissions than other liquid fertilizers and can be dozed more precisely [18]. The current study is part of a fine-tuning work with N2 Applied, where we examine different NEO types and compare their effects with mineral fertilizers and untreated slurry. A particular focus is on the effects of fertilizers on soil biological activity.

Theoretically, fertilizers improve soil biological activity through increased crop production and crop residue return [10]. On the other hand, N fertilizers, particularly ammonium-N, may increase soil acidy, leading to reduced biological activity or shifts in the functional groups in the soil [10,20,21]. Moreover, repeated applications could theoretically suppress certain soil enzymes involved in nutrient cycles, e.g., the amidase involved in the N cycle [20]. Studies show diminished soil microbial biomass under high N application rates in grassland, presumably through reduced plant species richness. A similar pattern has not been noticed under annual crop cultivations [9,22,23]. Nevertheless, the way soil functional groups confront recurrent mineral fertilization depends on different environmental and management factors [21].

Organic amendments, e.g., manure, compost, biosolids, and humic substances, serve as the carbon source for soil organisms, directly by their carbon content and indirectly via improved plant growth and plant residue returns. Overall, recurrent applications of organic fertilizers have promoted soil microbial growth and activity [24] and enhanced plant productivity [9]. The impacts of organic fertilizers on the microbial communities are more favorable than those of chemical fertilizers and vary between annual and perennial production systems [24]. Regarding soil fauna and invertebrates, their vulnerability to high N application rates differs. Detrimental effects on soil fauna feeding activity have been reported in both the short term [9,20] and long term [25]. However, other studies showed positive effects of fertilization on soil fauna composition and diversity and the feeding activity of the organisms [22,23,25]. These positive effects were evident in springtails in the topsoil [26]. Nevertheless, these controversies demonstrate the importance of more research to understand the changes in soil biota after fertilizers.

The current study investigates the effects of different fertilizers, including mineral fertilizer, NEO, and combinations of organic and mineral fertilizers, on ryegrass yields and soil fauna feeding activity under controlled conditions. The aims are: (1) to identify a fertilizer regime with optimal yields and compare NEOs' effect on yields to mineral fertilizer and organic fertilizer; and (2) to investigate the impact of the fertilizers on the soil fauna feeding activity. Here, NEO, as a new fertilizer, is our primary target.

## 2. Materials and Methods

### 2.1. Experiment Design

Pot experiments were conducted in a growing chamber at Inland Norway University of Applied Sciences. We used perforated pots (13 × 13 × 18 cm) filled with 2.5 L of field soil and fertilized them simultaneously. Each fertilizing treatment was randomized within four replicates.

Two subsequent trials were performed. Trial one consisted of 14 fertilizing treatments, and trial two was somewhat simplified with 11 to investigate and compare the effects of lower and more adjacent amounts of mineral fertilizer to the NEO (Table 1). The treatments were five fertilizing regimes; no fertilizer; different amounts of mineral fertilizer (Yara Mila 18-3-15) [27]; three types of NEO (N2 Applied) [17]; organic fertilizer (untreated cattle slurry); and organic fertilizer + different amounts of N in mineral fertilizer (Yara Liva 16-0-0) [28]. The mineral fertilizer levels were designed as a ladder from no fertilizer to a maximum of 235 kg N ha$^{-1}$ in trial one and 175 kg N ha$^{-1}$ in trial two. The Yara mineral fertilizers were added in the solid form; however, untreated slurry and all NEO types were added in liquid form. Treatments in trial one were (1) No fertilizer, (2) Mineral fertilizer 115 kg N ha$^{-1}$, (3) Mineral fertilizer 145 kg N ha$^{-1}$, (4) Mineral fertilizer 175 kg N ha$^{-1}$, (5) Mineral fertilizer 205 kg N ha$^{-1}$, (6) Mineral fertilizer 235 kg N ha$^{-1}$, (7) NEO type A 175.4 kg N ha$^{-1}$, (8) NEO type B 175.4 kg N ha$^{-1}$, (9) NEO type C 175.4 kg N ha$^{-1}$, (10) Organic fertilizer 73 kg N ha$^{-1}$, (11) Organic fertilizer + MF 115 kg N ha$^{-1}$, (12) Organic fertilizer + MF 145 kg N ha$^{-1}$, (13) Organic fertilizer + MF 175 kg N ha$^{-1}$, and (14) Organic fertilizer + MF 205 kg N ha$^{-1}$. Treatments in trial two were (1) No fertilizer, (2) Mineral fertilizer 60 kg N ha$^{-1}$, (3) Mineral fertilizer 80 kg N ha$^{-1}$, (4) Mineral fertilizer 115 kg N ha$^{-1}$, (5) Mineral fertilizer 135 kg N ha$^{-1}$, (6) Mineral fertilizer 155 kg N ha$^{-1}$, (7) Mineral fertilizer 175 kg N ha$^{-1}$ (8) NEO type A 175.4 kg N ha$^{-1}$, (9) NEO type B 175.4 kg N ha$^{-1}$, (10) NEO type C 175.4 kg N ha$^{-1}$, and (11) Organic fertilizer 73 kg N ha$^{-1}$.

Furthermore, the three types of NEO were different in terms of acidity, nitrate, and nitrite contents. The pH of NEO type A was 5.42 and contained 1530 mg L$^{-1}$ NH$_4^+$, 800 mg/L NO$_2^-$ and 1180 mg L$^{-1}$ NO$_3^-$. NEO type B had pH 5.35 and contained 1480 L$^{-1}$ NH$_4^+$, 777 L$^{-1}$ NO$_2^-$, and 1250 L$^{-1}$ NO$_3^-$. NEO type C had pH 4.24 and contained 1100 L$^{-1}$ NH$_4^+$, 444 L$^{-1}$ NO$_2^-$ and 1910 L$^{-1}$ NO$_3^-$. Moreover, the cattle slurry used in this experiment had pH 7.13, and it contained 1320 L$^{-1}$ NH$_4^+$. The target untreated manure amount was 55 tons ha$^{-1}$. The N2 Applied unit filters all particles larger than 5 mm during the production process, which reduces the total amount by 10 percent, down to 50 tons ha$^{-1}$. The untreated manure had 1.33 kg plant-available N per ton. However, after processing by the N2-Applied unit, the final product (NEO) had a plant-available N content of 3.51 kg per ton.

We used soil taken from the experimental farm. According to the soil analysis report from Eurofins (https://www.eurofins.no/agro-testing/ (accessed on 30 January 2022)) on samples taken prior to the experiment, the soil texture was sandy clay loam with over 10% clay and 4.5% soil organic matter. The soil pH was high (pH = 7.4) while the phosphorus status was normal (P-AL = 11 mg/100 g), and the potassium status was low (K-AL = 5 mg/100 g). The field water capacity was estimated to be 33.6% of soil volume and with a pore capacity of 41.4%.

Pot preparation and planting were done following the procedure that we developed. First, a moistened paper tissue was placed at the bottom of the perforated pots to avoid soil eruption through the holes. Then, 0.6 L (5 cm) soil was filled into the pots. The second layer, 0.8 L (6 cm) soil, was fertilized and filled into the pots. Fertilizer amounts were based on the recommended field application rates (in tons per hectare), calculated for 169 cm$^2$ soil surface per pot (Table 1). Next, an additional 0.9 L (6 cm) soil was filled in the pots, and seeds were placed over the top. Thirty-six Italian ryegrass seeds (*Lolium multiflorum* Lam.), variety 'Barpluto' (NAK Nederland/Ref. DE148-214011) were seeded with 12 seeds in three rows per pot. Then additional 0.2 L (1 cm) soil was filled on the top to form the outermost soil layer.

We used a Lumatek ATS300W LED lighting system (https://lumatek-lighting.com/ (accessed on 30 January 2022)). The LED light pads provided a full spectrum of light (380–780 nm wavelength) suitable for growing cereals and grasses under controlled conditions [29]. For this experiment, five LED pads were positioned in a row and 35 cm over the top foliage. The LED pads were lifted as the plants grew. According to a typical Nordic summer day length, the light/dark duration was set to 16 h light and 8 h dark. Moreover,

light intensity was measured using a digital light meter for approving equal light access for 16 pots per lamp (80 × 80 cm). The temperature in the growing chamber was 16 °C during this experiment.

**Table 1.** Fertilizing treatments used in trials one and two with the different fertilization treatments in different colors and detailed application rates.

| Trial One | Fertilizing Treatment | Organic Fertilizer (tons ha$^{-1}$) | Kg N in Yara Mila18-3-15 (kg ha$^{-1}$) | Kg N in Organic Fertilizer (kg ha$^{-1}$) | Kg N in Yara Liva 16-0-0 (kg ha$^{-1}$) | Total kg N (kg ha$^{-1}$) |
|---|---|---|---|---|---|---|
| 1 | No fertilizer | - | - | - | - | 0 |
| 2 | Mineral fertilizer 115 kg N ha$^{-1}$ | | 115 | | | 115 |
| 3 | Mineral fertilizer 145 kg N ha$^{-1}$ | | 145 | | | 145 |
| 4 | Mineral fertilizer 175 kg N ha$^{-1}$ | | 175 | | | 175 |
| 5 | Mineral fertilizer 205 kg N ha$^{-1}$ | | 205 | | | 205 |
| 6 | Mineral fertilizer 235 kg N ha$^{-1}$ | | 235 | | | 235 |
| 7 | NEO type A 175.4 kg N ha$^{-1}$ | 50 | | 175.4 | | 175.4 |
| 8 | NEO type B 175.4 kg N ha$^{-1}$ | 50 | | 175.4 | | 175.4 |
| 9 | NEO type C 175.4 kg N ha$^{-1}$ | 50 | | 175.4 | | 175.4 |
| 10 | Organic fertilizer 73 kg N ha$^{-1}$ | 55 | | 73 | | 73 |
| 11 | Organic fertilizer + MF 115 kg N ha$^{-1}$ | 55 | | 73 | 42 | 115 |
| 12 | Organic fertilizer + MF 145 kg N ha$^{-1}$ | 55 | | 73 | 72 | 145 |
| 13 | Organic fertilizer + MF 175 kg N ha$^{-1}$ | 55 | | 73 | 102 | 175 |
| 14 | Organic fertilizer + MF 205 kg N ha$^{-1}$ | 55 | | 73 | 132 | 205 |
| **Trial Two** | **Fertilizing Treatment** | **Organic Fertilizer (tons ha$^{-1}$)** | **Kg N in Yara Mila18-3-15 (kg ha$^{-1}$)** | **Kg N in Organic Fertilizer (kg ha$^{-1}$)** | **Kg N in Yara Liva 16-0-0 (kg ha$^{-1}$)** | **Total kg N (kg ha$^{-1}$)** |
| 1 | No fertilizer | - | - | - | - | 0 |
| 2 | Mineral fertilizer 60 kg N ha$^{-1}$ | | 60 | | | 60 |
| 3 | Mineral fertilizer 80 kg N ha$^{-1}$ | | 80 | | | 80 |
| 4 | Mineral fertilizer 115 kg N ha$^{-1}$ | | 115 | | | 115 |
| 5 | Mineral fertilizer 135 kg N ha$^{-1}$ | | 135 | | | 135 |
| 6 | Mineral fertilizer 155 kg N ha$^{-1}$ | | 155 | | | 155 |
| 7 | Mineral fertilizer 175 kg N ha$^{-1}$ | | 175 | | | 175 |
| 8 | NEO type A 175.4 kg N ha$^{-1}$ | 50 | | 175.4 | | 175.4 |
| 9 | NEO type B 175.4 kg N ha$^{-1}$ | 50 | | 175.4 | | 175.4 |
| 10 | NEO type C 175.4 kg N ha$^{-1}$ | 50 | | 175.4 | | 175.4 |
| 11 | Organic fertilizer 73 kg N ha$^{-1}$ | 55 | | 73 | | 73 |

### 2.2. Growing Conditions and Yield

Primary irrigation was done with 500 mL water to provide adequate moisture. This amount was 55% of field capacity for our dry soil, whereas the soil was not wholly dry

at planting. During plants' growth, the irrigation routine was 200 mL water three times a week for the first weeks. Irrigation intervals were 200 mL every 1–2 days during the last weeks before harvest progressing the plant developmental stages [30].

The pots were placed adhering to each other for the first two weeks. A five cm distance was applied between pots from week three to prevent plants from competing for light and space. After germination, the least vigor plants in each row were thinned, allowing 24 plants per pot (eight plants per row). A few weeds germinated per pot during the experiment, and these were removed by hand.

In trial one, plants were harvested six weeks after planting. Moreover, a second harvest took place three weeks after the first harvest. Harvesting was done by cutting the plants 1 cm from the soil surface. Bulk yield and dry matter yield (DM) from each pot were measured at harvest and after drying at 60 °C for 48 h [31]. In trial two, a single harvest was done eight weeks after planting. Otherwise, the same procedure was followed as described for trial one.

### 2.3. Feeding Activity of Soil Fauna

The soil fauna feeding activity was assessed using Bait-lamina strips (Terra Protecta GmbH, Berlin, Germany) [32]. The Bait-lamina test is an efficient, prompt, and replicable method with high statistical relevance via several replications [33,34]. This method evaluates soil fauna feeding activity as a decisive function in nutrient cycling [34–36]. The method has revealed promising results regarding the soil fauna feeding activity for screening and comparing them under different managements and practices [34]. The method exposes perforated PVC strips (1 mm × 6 mm × 120 mm) with 16 holes of 1.5 mm diameter with 5 mm distance filled with a bait substrate to soil fauna (invertebrates) feeding activity. The substrate consists of 70% cellulose powder, 25% wheat bran, and 5% activated carbon [37]. The loss of substrate after a certain period indicates soil fauna feeding activity, while soil flora (e.g., bacteria, fungi, etc.) play a minor role [38–40].

In trial one, concurrently with the first harvest (six weeks after planting/fertilization), three diametrical Bait-lamina strips were inserted into each pot (replicate) with the uppermost hole below the soil surface for assessing the early effects of fertilization on soil fauna feeding activity [37]. In addition, another set of strips was inserted 21 weeks after planting/fertilization to determine the late effects. In trial two, the strips were inserted into the soil after the harvest (eight weeks after planting/fertilization). Plant growth and irrigation continued as usual for the whole period that strips were in the soil. The strips were exposed to soil fauna feeding activity for seven weeks on each occasion before the strips were removed and visually inspected for the loss of the bait substrate [37]. The loss of substrate in any hole per strip was scored as empty (1), partly empty (0.5), or filled (0) [9]. Every unfilled hole (score 1) was equal to 6.25% feeding activity at any defined depth.

### 2.4. Statistical Analyzes

The data were registered and cleaned. The differences in yield and soil fauna feeding activity were analyzed using Minitab 20 statistical software (© 2021 Minitab, LLC (State College, PA, USA)). A one-way ANOVA test was used to evaluate the differences between fertilizing treatments. In addition, Games–Howell pairwise comparison was used to compare and group differences between treatments and plot data at a 95% confidence interval for the means. Individual standard deviations are used to estimate the confidence intervals.

## 3. Results

### 3.1. Dry Matter Yields

Dry matter (DM) yield in trial one reflects the N content provided in fertilization, and a significant difference ($p = 0.001$) was observed between the different fertilizing treatments (Figure 1A, Table S1). Summing up the DM yields from both first and second harvests showed that the highest amount of mineral fertilizer (235 kg N ha$^{-1}$) produced between 12–46% more yields than other fertilization treatments and 81% more yields than

no fertilizer. Moreover, NEO yielded at a higher level than untreated manure and in the same range as 175 kg N ha$^{-1}$ in mineral fertilizer, which was the same N content as in the NEO. It was also evident that no fertilizer had the lowest DM yields (Figure 1A, Table S1).

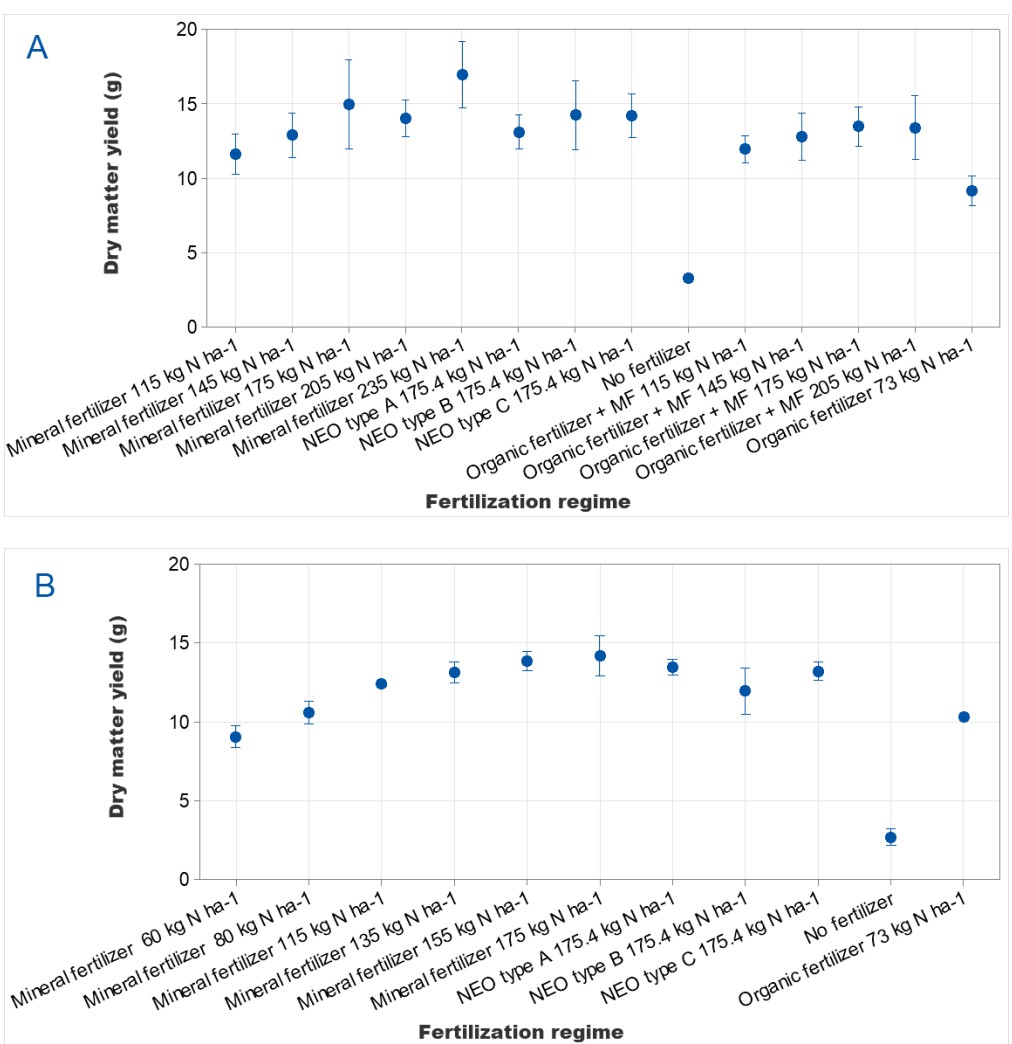

**Figure 1.** Effects of different fertilizing treatments on ryegrass dry matter yields (g) in trial one (**A**) and trial two (**B**). Individual standard deviations at 95% confidence interval are used in the graphs.

Similar to trial one, in trial two, the N amount in the fertilizer significantly affected the DM yields ($p = 0.001$) (Figure 1B, Table S1). Mineral fertilizer 175 kg N ha$^{-1}$ and 155 kg N ha$^{-1}$ had higher DM yields than the other treatments. Mineral fertilizer 175 kg N ha$^{-1}$ produced between 3 and 63% more yield than other fertilization treatments and 81% more than no fertilizer. The different NEO fertilizers produced in the same range as mineral fertilizers with 115–155 kg N ha$^{-1}$ while mineral fertilizer 80 kg N ha$^{-1}$, organic fertilizer 73 kg N/ha$^{-1}$, and mineral fertilizer 60 kg N ha$^{-1}$ had lower DM yields than the other treatments with no fertilizer in the bottom (Figure 1B, Table S1).

### 3.2. Soil Fauna Feeding Activity

In trial one, soil fauna feeding activity was assessed for early and late fertilization effects. There was a significant difference between different fertilizing treatments ($p = 0.001$) regarding the early effects in trial one. Mineral fertilizer 205 kg N ha$^{-1}$ had a higher soil fauna feeding activity than mineral fertilizer 115 kg N ha$^{-1}$, NEO type B 175.4 kg N ha$^{-1}$, and NEO type C 175.4 kg N ha$^{-1}$ (Figure 2A, Table S1).

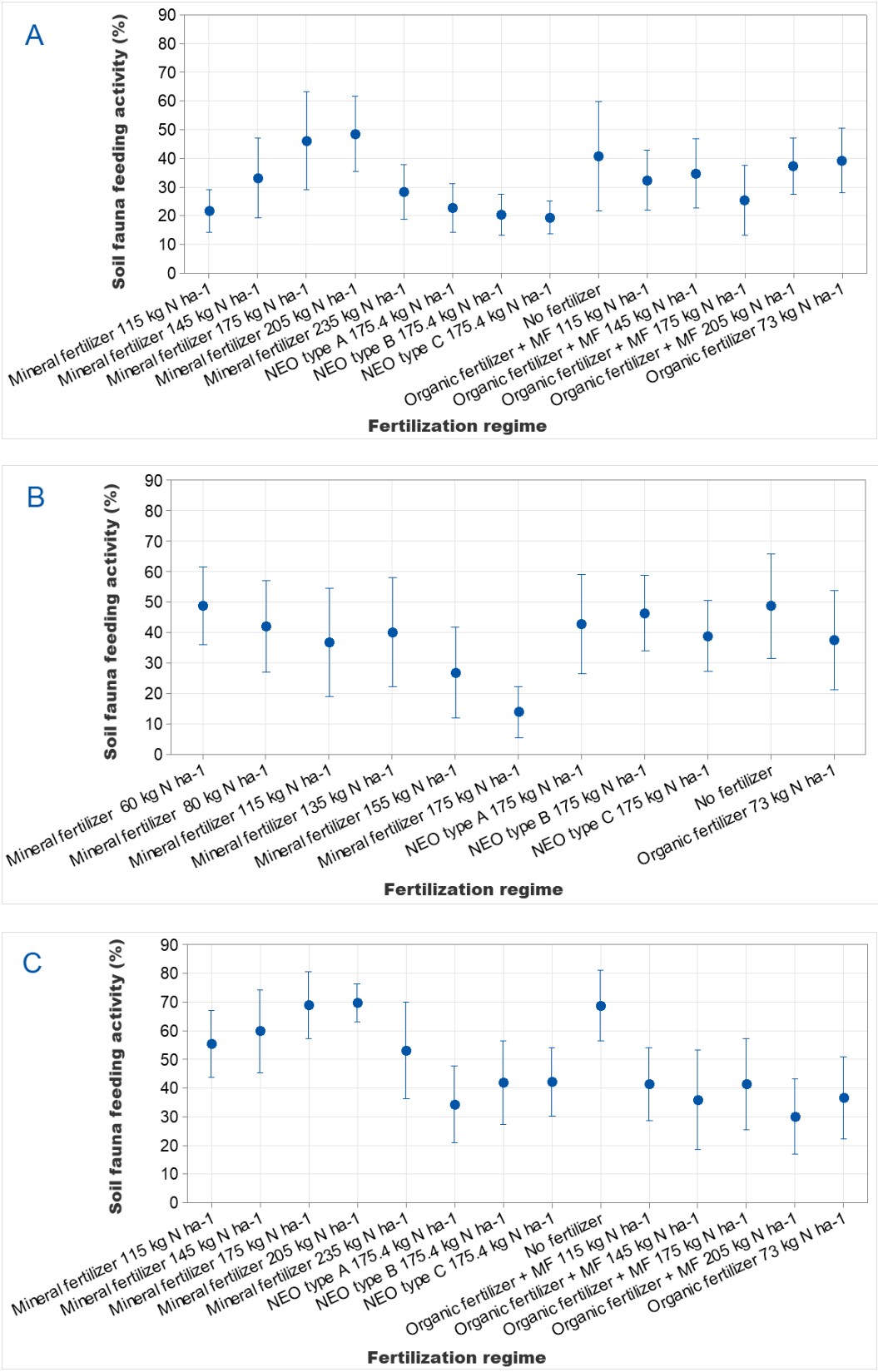

**Figure 2.** Effects of different fertilizing treatments on soil fauna feeding activity (%) in the early effect measurement in trial one (**A**) and trial two (**B**), and in the late effect measurement in trial one (**C**). Individual standard deviations at 95% confidence interval are used in the graphs.

　　　Also in trial two, there was a significant difference between different fertilizing treatments ($p = 0.001$) regarding the early effects. Here, no fertilizer, NEO type B 175.4 kg N ha$^{-1}$, and mineral fertilizer 60 kg N ha$^{-1}$ had higher soil fauna feeding activity than other treatments. However, mineral fertilizer 175 kg N ha$^{-1}$ had the lowest soil fauna feeding activity (Figure 2B, Table S1).

　　　Regarding the late effects of fertilizers on soil fauna feeding activity in trial one, likewise the early effects, there was a significant difference between different fertilizing treatments ($p = 0.001$). Mineral fertilizer 205 kg N ha$^{-1}$ had a higher soil fauna feeding activity than organic fertilizer 73 kg N ha$^{-1}$, all types of NEO 175.4 kg N ha$^{-1}$, and all combinations of organic fertilizer + mineral fertilizer. NEO type A 175.4 kg N ha$^{-1}$ and organic fertilizer + mineral fertilizer 205 kg N ha$^{-1}$ had the lowest soil fauna feeding activity (Figure 2C, Table S1).

　　　We identified a pattern where the highest feeding activity occurred in depth below 5 cm from the soil surface. In trial one, the feeding activity was at its lowest at 2 cm depth ($p = 0.001$). In trial two, there was no significant difference in the depth of feeding activity ($p = 0.08$). Both these results refer to the early effects after fertilization. The same pattern was observed for the late fertilization effects, but the differences were not significant ($p = 0.37$) (Figure 3A–C, Table S1).

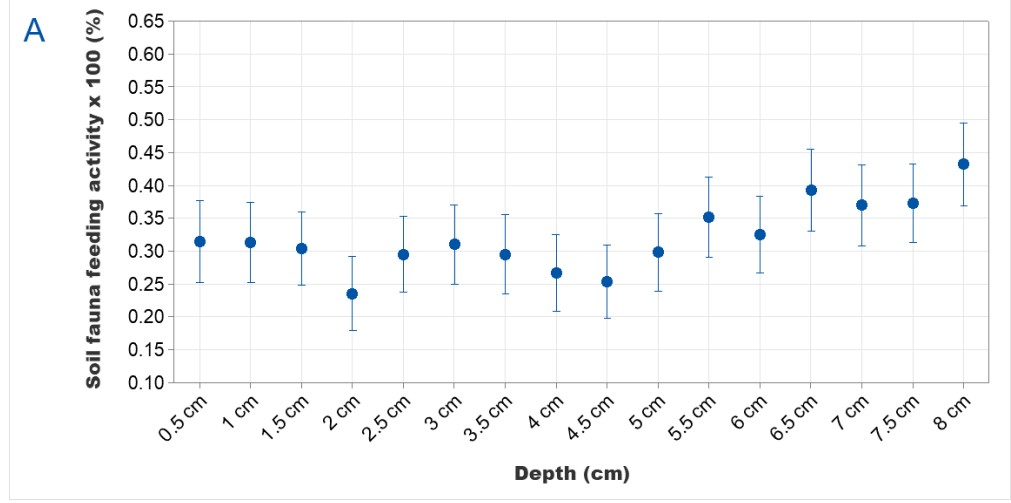

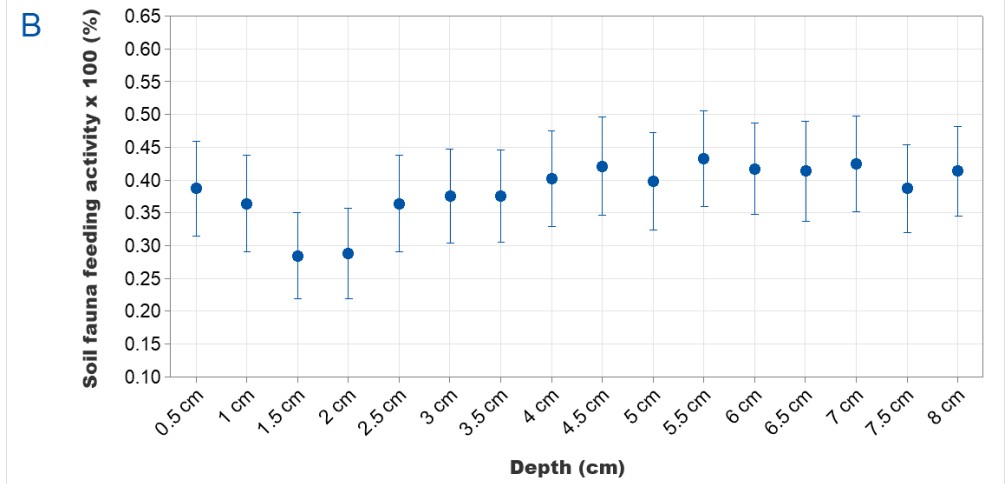

**Figure 3.** *Cont.*

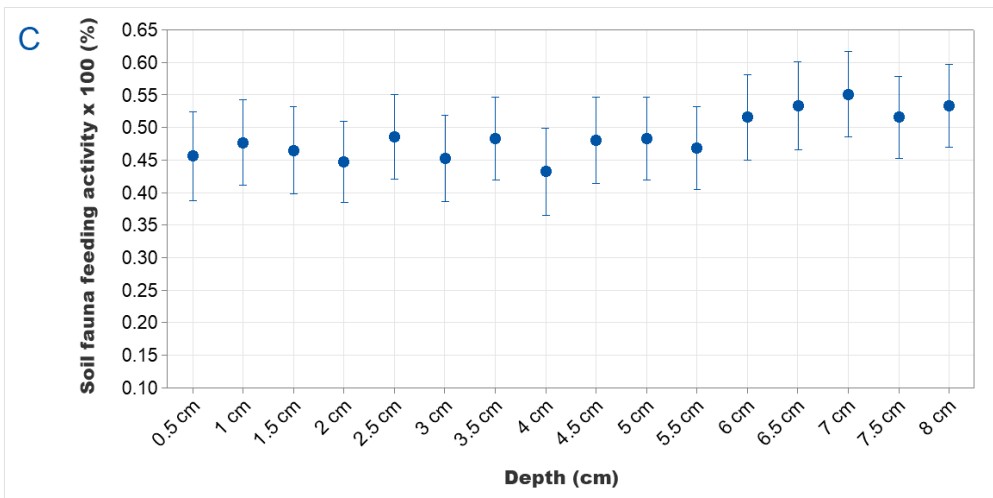

**Figure 3.** Effects of different fertilizing treatments on the depth of soil fauna feeding activity ($\times$100%) in the early effect measurement in trial one (**A**) and trial two (**B**), and the late effect measurement in trial one (**C**). Individual standard deviations at 95% confidence interval are used in the graphs.

## 4. Discussion

NEO is a newly patented nitrogen enriched organic fertilizer and is considered a sustainable product. However, its effect on plant yield and soil living organisms has not yet been examined. Here, we investigated the alteration of ryegrass yields and soil fauna feeding activity on a sandy clay loam where NEO was compared to mineral and organic fertilizers under different fertilizing levels under controlled conditions in pot experiments.

Overall, our results showed, as expected, a positive effect of nitrogen on ryegrass yields where the yield increased with increased mineral fertilization. The N ladder in mineral fertilizer application was designed to judge the fertilization value of NEO. Application of NEO with 175 kg N ha$^{-1}$ yielded in the same range or a little lower than mineral fertilizers with the same N content. This was a positive result for NEO. An organic N fraction can explain the slightly lower yield in the NEO (like in manure), which needs to be mineralized before it is available for plants. Furthermore, we showed that NEO produced 31–36% higher yields than non-treated manure. There were no differences in yields among the three NEO types, even though they differed in the percentage of nitrite and nitrate but with the same total N content.

The positive effect of mineral nitrogen and ryegrass yield has been documented before [6–8]. We used a sandy clay loam and found the highest yields at our maximum level of 235 kg N ha$^{-1}$ in trial one and 175 kg N ha$^{-1}$ in trial two. A similar result was found in field trials on a clay loam in South Africa with the highest yields at 240 kg N ha$^{-1}$ [8], which was the maximum fertilization level in that study. The same result was reported on clay soils in Turkey [8]. Harris et al. [6] demonstrated increasing yields up to 400 kg N ha$^{-1}$ on different loam soils in New Zealand, which is a much higher nitrogen level than we used. We could not detect a superior effect of mineral and organic fertilizer combined, as described by others [5].

While the relationship between fertilization and plant yields is well established, the knowledge about its impacts on soil-living communities requires improvements [41], and especially the knowledge on NEO as a new product is very limited. In the early effect measurement (with the Bait-lamina strips put into the soil six weeks after fertilization), fertilizers in general tend to diminish the soil fauna's feeding activity more than no fertilizer. However, there was no indication of any severe adverse impacts on the soil fauna feeding activity. This is exemplified in trial one, where mineral fertilizers with 175 kg N ha$^{-1}$ imposed higher soil fauna feeding activity than no fertilizer. Nevertheless, this phenomenon was not verified in trial two. In trial one, the treatment with the lowest amount of mineral fertilizer (115 kg N ha$^{-1}$) and NEO types B and C had the lowest soil fauna feeding activity

in early effect measurement, but this pattern was not confirmed in trial two. Instead, in trial two, all NEO types, organic fertilizer, and the lowest amounts of mineral fertilizer (60 and 80 kg N ha$^{-1}$) had better or similar soil fauna feeding activity as the highest amounts of mineral fertilizer. Taken together, all fertilizing treatments, NEO included, showed a trend with lower feeding activity compared to no fertilizer. The exceptions to this trend were mineral fertilizer 175 kg N ha$^{-1}$ and mineral fertilizer 205 kg N ha$^{-1}$ in trial one. Regarding the late effects measured after 21 weeks, fertilization diminished soil fauna feeding activity compared to no fertilizer regardless of fertilizer type. Again, mineral fertilizer 175 kg N ha$^{-1}$ and 205 kg N ha$^{-1}$ were exceptions to this trend, and organic fertilizer, all organic fertilizer + mineral fertilizer combinations, and all types of NEO showed a lower percentage of soil fauna feeding activity than no fertilizer. Thus, we see a pattern where fertilization, including NEO, tends to decrease the soil fauna feeding activity. Other studies also indicate adverse effects of mineral fertilizers on soil fauna, validating this argument [9,42–44]. However, the other valid argument is increased food availability for the soil fauna through fertilizers, whereas the soil fauna has no interest in consuming bait material in the strips. Contrasting the outcomes of this study, field studies have indicated positive effects of mineral fertilizer on soil fauna, presumably through increasing nutrient availability and enhancing plant productivity [45,46], and organic fertilizer through their advantages for plant production systems [47,48]. Therefore, we argue that other factors than fertilization alone may explain our result and affect the result. Considering that the observed tendencies partly contradict, we could conclude that there is neither a positive nor a negative effect of the fertilizing treatment on soil fauna feeding activity. One last outcome was that the soil fauna feeding activity was higher in the late measurement (21 weeks after fertilization) than in the early measurement (six weeks after fertilization). Therefore, it is valid to argue that with time, plants' root network is expanding in the soil, providing pathways for more convenient movement of soil fauna [49]. Developing a rhizosphere is a time-demanding process, especially in a closed system such as a pot experiment. We also saw that the feeding activity increased in depth below 5 cm from the soil surface. This conflicts with the finding of most field studies, where the most soil fauna feeding activity takes place in the top 5 cm of soil surface [34]. However, arguably in a pot trial where the irrigation occurs over the soil surface, nutrients and the plant roots accumulate in the pot's bottom. It is also feasible to interpret that we mixed fertilizers with the second soil layer from the bottom of the pot, which induced more feeding activity within this layer. These arguments are presumably the driving forces for more soil fauna feeding activity at this depth.

## 5. Conclusions

Nitrogen Enriched Organic fertilizer (NEO) produced ryegrass yields in the same range as mineral fertilizer with similar total nitrogen content, and NEO produced clearly higher yields than untreated organic manure. There was no correlation between yield data and soil fauna feeding activity. NEO, as the other fertilizer treatments, had some negative effects on soil fauna feeding activity compared to no fertilizer that tends to have the highest feeding activity. The variation was, however, big and factors other than fertilization alone seem to influence the soil fauna feeding activity. Alternatively, the Bait-lamina method is not applicable for this type of pot experiment under controlled conditions. More research is needed to clarify these uncertainties. Therefore, as the next step, we are progressing with field trials and biochemical analyses of the soil samples to investigate the effects of NEO on plant yield and soil organisms under field conditions.

**Supplementary Materials:** Supporting information can be downloaded from https://www.mdpi.com/article/10.3390/su14042005/s1, Table S1: Effects of different fertilizing regimes including different amounts of mineral fertilizer, three types of NEO, organic fertilizer (untreated cattle slurry), organic fertilizer + different amounts of mineral fertilizer (MF), and no fertilizer on ryegrass dry matter yields (g) in trial one (A) and trial two (B), soil fauna feeding activity (%) in the early effects in trial one (C) and trial two (D), and in the late effects in trial one (E), and the depth of feeding activity

(%) in the short term in trial one (F) and trial two (G), and in long term trial one (H). Games-Howell pairwise comparison method at 95% confidence interval is used to compare the differences between means. Means that do not share a letter are significantly different.

**Author Contributions:** Conceptualization, H.M. and T.C.; methodology, H.M. and T.C.; validation, H.M. and S.Ø.S.; formal analysis, H.M.; investigation, H.M. and T.C.; lab work and data acquisition: H.M. and G.H.; data curation, H.M.; writing—original draft preparation, H.M. and S.Ø.S.; writing—review and editing, H.M., S.Ø.S. and T.C.; visualization, H.M.; supervision, S.Ø.S.; project administration, T.C.; funding acquisition, T.C. and S.Ø.S. All authors have read and agreed to the published version of the manuscript.

**Funding:** This research was funded by the Research Council of Norway.

**Institutional Review Board Statement:** Not applicable.

**Informed Consent Statement:** Not applicable.

**Data Availability Statement:** Data is accessible at https://osf.io/deukp/?view_only=ecb72edabb4 24f5cb38c11a40773d012 (accessed on 30 January 2022).

**Acknowledgments:** The authors would like to thank Morten Tofastrud and Elisabeth Røe at Inland Norway University of Applied Sciences for support in project administration, and in addition, we would like to acknowledge the project's funders.

**Conflicts of Interest:** The authors declare no conflict of interest.

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
