# Peer review of "Nitrogen Enriched Organic Fertilizer (NEO) and Its Effect on Ryegrass Yield and Soil Fauna Feeding Activity under Controlled Conditions"

_sustainability, doi:10.3390/su14042005_

Round 1
Reviewer 1 Report
1- Please use scientific name of Italian ryegrass in the abstract section.
2- please remove at least from line 13 in abstract section.
3- please mention treatments with numbers and clearly mention in the abstract and M&M section.
4- Objectives should be in numbers. (i), (ii) like this for clear understanding. 5- L92. remove pot from pot replicate soil.
6-Have you applied fertilizers in liquid form or in solid form in each experimental unit.
7- Please summarize this heading "Plant growth conditions and yield". measurements
8- Please clearly mention the Y-axis of all the figures and improve the figures quality.
9- If possible, please add more reasons in the discussion portion to support your novel findings.
10- Authors should add some information regarding their key findings in values in any unit.
Author Response
Please kindly find the response to your concerns regarding the manuscript in the attached word file.

Reviewer 2 Report
This study explored the effects of a new organic fertilizer NEO on Ryegrass yield and soil fauna feeding activity. The research method is relatively simple and the content is not sufficient. The specific opinions are as follows:
- The results showed that ryegrass yield increased with the increase of mineral fertilization, which was not innovative and of little significance.
- “The NEO fertilizers had no adverse effects on soil fauna feeding activity than other fertilizer treatments. ”The results in Figure 2 (C) seemed to contradict this conclusion.
- "but the no fertilizer treatment tends to have the highest feeding activity. Thus, factors other than fertilization seem to influence the soil fauna feeding activity. "I personally believe that the soil fauna feeding activity without fertilization is the highest, indicating that fertilization may be detrimental to the activities and richness of animals, and fertilization will affect the soil fauna feeding activity. The conclusion is not accurate enough.
- “but we could not detect a positive effect of a combination of mineral and organic fertilizer as described by Körschens, Albert, Armbruster, Barkusky, Baumecker, Behle-Schalk, Bischoff, ÄŒergan, Ellmer, Herbst, Hoffmann, Hofmann, Kismanyoky, Kubat, Kunzova, Lopez-Fando, Merbach, Merbach, Pardor, Rogasik, Rühlmann, Spiegel, Schulz, Tajnsek, Toth, Wegener and Zorn” Names can be changed to “Körschens, Albert, Armbruster, etc.”
- This research can be said to be a preliminary test of NEO, the research content is not innovative enough, and the yield research with pot experiments is not rigorous enough.
- Some studies on soil physicochemical properties and microbial communities can be added to clarify the application effect of NEO in agricultural practices.
Author Response

(The authors gave the same response as above.)

Reviewer 3 Report
These research findings are good in terms of scaling down the research finding by using the potting technique. However, from the data collection, it is good to compare to other researches findings, to give more accurate and helpful in discussion and conclusion section.

Author Response
Dear reviewer,
First of all, thank you for your positive feedback regarding our manuscript. You can find the response to your specific comments in the attached PDF file. Moreover, the discussion and conclusion sections of the manuscript underwent major changes. I hope the changes could be satisfactory according to your standards.
Kind regards,
On behalf of the authors,
Hesam Mousavi

Round 2
Reviewer 2 Report
We looking forward to the application effect of NEO with field trials and related analysis results.